# Impact of Prior SARS-CoV-2 Infection on COVID-19 Vaccine Effectiveness in Children and Adolescents in Norway and Italy

**DOI:** 10.3390/vaccines13070698

**Published:** 2025-06-27

**Authors:** Elisa Barbieri, Nhung T. H. Trinh, Costanza Di Chiara, Giovanni Corrao, Riccardo Boracchini, Ester Rosa, Cecilia Liberati, Daniele Donà, Angela Lupattelli, Carlo Giaquinto, Anna Cantarutti

**Affiliations:** 1Department for Women’s and Children’s Health, University of Padua, 35128 Padua, Italy; 2Pharmacoepidemiology and Drug Safety Research Group, Department of Pharmacy, Faculty of Mathematics and Natural Sciences, University of Oslo, 0316 Oslo, Norway; 3Penta—Child Health Research, 35127 Padua, Italy; 4National Centre for Healthcare Research and Pharmacoepidemiology, University of Milano-Bicocca, 20126 Milan, Italy; 5Unit of Biostatistics, Epidemiology and Public Health, Department of Statistics and Quantitative Methods, University of Milano-Bicocca, 20126 Milan, Italy; 6Società Servizi Telematici-Pedianet, 35138 Padova, Italy

**Keywords:** mRNA vaccination, hybrid immunity, COVID-19, pediatric population, vaccine effectiveness

## Abstract

**Background and objective**: The approval of mRNA-based vaccines for children and adolescents has contributed to global efforts to control the SARS-CoV-2 pandemic. While hybrid immunity—combining prior SARS-CoV-2 infection and vaccination—may offer enhanced protection, data on its effectiveness versus vaccine-induced immunity in the pediatric population are limited. **Methods**: This retrospective matched cohort study used linked health data from Norwegian nationwide health registries and the Italian Pedianet network. The study included children and adolescents aged 5–14 years eligible for COVID-19 vaccination at the time of approval (May/September 2021 and November 2021/January 2022, respectively). Mono- and two-dose vaccination schedules were assessed, and hybrid immunity was defined as prior SARS-CoV-2 infection followed by vaccination within 12 months. Conditional Cox regression models were used to estimate hazard ratios (HRs) for SARS-CoV-2 infection risk, adjusting for sociodemographics, comorbidities, and healthcare utilization. **Results**: The study included 626,537 children and adolescents in Norway and 38,938 in Italy. A single dose of the vaccine did not reduce the risk of infection among SARS-CoV-2–naive individuals in Norway (HR: 1.05; 95% CI: 1.04–1.07), whereas it was associated with an 8% risk reduction in Italy (HR: 0.92; 95% CI: 0.88–0.96). Among individuals with a recent prior infection (within 12 months), vaccination was associated with a reduced risk of reinfection in Norway (HR: 0.10; 95% CI: 0.05–0.13), but not in Italy (HR: 1.22; 95% CI: 0.83–1.80), compared to no vaccination. Among those with prior infection, vaccination was associated with a significantly reduced risk of reinfection in Norway (HR = 0.10; 95% CI: 0.05–0.20), but not in Italy (HR = 0.55; 95% CI: 0.27–1.11). Hybrid immunity provided greater protection against (re-)infection compared to vaccine-induced immunity alone, with a 26% risk reduction observed in Norway (HR = 0.74; 95% CI = 0.47–0.1.16) and an 86% reduction in Italy (HR = 0.14; 95% CI = 0.09–0.21). **Conclusions**: This analysis supports the effectiveness of SARS-CoV-2 vaccines in children, with hybrid immunity offering enhanced protection against reinfection. Given the waning effectiveness of vaccines over time, continued research and booster strategies are essential to sustain protection and mitigate transmission.

## 1. Introduction

In May and November 2021, the US Food and Drug Administration and the European Medicines Agency authorized the first mRNA-based vaccines against SARS-CoV-2—Tozinameran (Comirnaty, BioNTech–Pfizer, Mainz, Germany/New York, NY, USA) and Elasomeran (Spikevax, Moderna, Cambridge, MA, USA) [1,2]. Shortly after, these vaccines were introduced in national vaccination campaigns. In Italy, the rollout began for adolescents in May 2021 and for children in November 2021 [3]. Similarly, in Norway, adolescents were offered vaccination from September 2021 and children from January 2022. Despite official recommendations in Italy and established evidence supporting the safety and efficacy of these vaccines, uptake remained below optimal levels in both countries [4]. In Italy, 84% of adolescents (aged 12–19) and 35% of children (aged 5–11) received at least one dose [5,6]. In Norway, only 2.6% of children (aged 5–11) and 73.3% of adolescents (aged 12–15) received at least one dose [7].

Several factors may have contributed to the lower vaccine uptake, including parental uncertainty regarding the magnitude and durability of vaccine-induced immunity compared to the combined protection from natural infection and vaccination (i.e., hybrid immunity) [8,9].

While protection against SARS-CoV-2 infection has been primarily studied by comparing vaccinated versus unvaccinated individuals [10,11] and SARS-CoV-2-recovered versus SARS-CoV-2-naïve individuals [12,13], the following findings have highlighted the complex interplay between prior infection, vaccination, and circulating SARS-CoV-2 variants [14]. Reinfections have been shown to occur more rapidly during periods of Omicron variant dominance, with the effectiveness of prior natural immunity against Omicron differing based on the variant responsible for the prior infection. For example, immunity from a pre-Omicron Delta (B.1.617.2) infection was significantly less effective against reinfection with BA.4 and BA.5 subvariants compared to BA.1 and BA.2 [15].

Several studies, including a target trial emulation in Israel, have documented the superior effectiveness of hybrid immunity against reinfection with the Delta and Omicron variants compared to both vaccine-induced and natural immunity in children and adolescents. A single dose of tozinameran vaccine in previously infected children and adolescents provided 78% and 64% greater protection against the Delta variant and 54% and 71% greater protection against the Omicron BA.1 and BA.2 subvariants, respectively, compared to unvaccinated children and adolescents with prior infection. However, the vaccine did not confer substantial protection against the BA.4 and BA.5 Omicron subvariants [16]. Despite these findings, knowledge about the effectiveness of hybrid immunity in other populations and settings with varying infection dynamics remains limited. For instance, Norway experienced low infection rates, whereas Italy faced a high burden of infection and severe disease, particularly during the early phases of the pandemic.

In this study, we aimed (i) to assess the effectiveness of a single dose of COVID-19 vaccine, considering factors such as prior SARS-CoV-2 infection, the time elapsed since the last infection, and the specific variant of concern (VoC) (i.e., pre-Omicron or Omicron); and (ii) to compare the effectiveness of hybrid immunity (i.e., combined protection of naturally acquired and vaccine-induced immunity) versus complete vaccination (i.e., two doses administered within the recommended interval) in the prevention of SARS-CoV-2 infection in a cohort of children and adolescents aged 5–14 years in Norway and the Veneto Region, Italy.

## 2. Materials and Methods

### 2.1. Study Setting and Data Sources

This retrospective study used Norwegian-linked health registries and the Italian Pedianet network data.

The national health registries in Norway include linked data from the Norwegian Patient Registry (NPR), the Norway Control and Payment of Health Reimbursement (KUHR), the Norwegian Surveillance System for Communicable Disease (MSIS), and the Norwegian Immunization Registry (SYSVAK) [17]. Additionally, the data were linked with administrative data from Statistics Norway (SSB) using the unique personal identification number assigned to all residents in Norway. This approach ensures comprehensive coverage of the Norwegian population of approximately 5.4 million inhabitants and encompasses data from primary and secondary healthcare settings. These national health registries and SSB in the context of children and adolescent populations have been described in detail elsewhere [7].

Pedianet is a national population database that contains anonymous patient-level data of more than 500,000 children since 2004, corresponding to around 4% of the annual paediatric population in Italy [18]. For this study, we included data from 80 family paediatricians (FPs) from the Veneto Region who used the same software (JuniorBit 7—https://sosepe.com/ho-scelto-junior-bit/, accessed on 25 April 2025) in their professional practice and contributed to the database from 2004 up to 15 May 2023, when the data were extracted. The Pedianet database has previously been described in detail [18]. Briefly, Pedianet records patient-level information, including demographic data, health status, clinical symptoms, and outcomes. The Pedianet Veneto is also enhanced by the electronic health record of the Veneto Region, including the hospital discharges registry, which reports all diagnoses released from public hospitals; the SARS-CoV-2 swabs registry, which records details of all nasopharyngeal swabs carried out; and the immunization registry, which records detailed information on vaccination performed. Documentation of infection was based on positive PCR or rapid antigen tests. Methods for real-time, quantitative reverse-transcriptase PCR testing and rapid antigen testing at healthcare facilities were provided elsewhere [19]. Data are anonymized to a centralized database based in Società Servizi Pediatrici, the legal owner of Pedianet, in Padova, Italy.

### 2.2. Study Design and Participants

We conducted a retrospective matched cohort study separately for each country.

In Norway, we included children aged 5–11 years in January 2022 (the approval date of COVID-19 vaccination for this age group) and adolescents aged 12–14 years in September 2021 (the approval date for adolescents). In Italy, we included children aged 5–11 years who became eligible for COVID-19 vaccination in November 2021 and adolescents aged 12–14 who became eligible in May 2021. These children were followed by family physicians in the Pedianet Veneto network and adhered to recommended well-child visit schedules, as described in previous research (Appendix A) [20].

Children and adolescents who received COVID-19 vaccines prior to the approval dates in either country were excluded from the study.

### 2.3. Exposure Definitions

#### 2.3.1. Vaccine-Induced Immunity with One Dose

Children and adolescents were classified as exposed if they had received at least one dose of any mRNA COVID-19 vaccine. Exposed individuals were matched 1:1 to unexposed children or adolescents, randomly selected from the target population, who had not yet been vaccinated at the index date. The index date was defined as the date of the first COVID-19 vaccine dose. Matching was applied based on age class at the month of vaccine approval dates (i.e., January 2022 and November 2021 for children, September 2021 and May 2021 for adolescents, in Norway and Italy, respectively), sex, and proxies of individual and familial sociodemographics. These proxies included parental COVID-19 vaccination status in Norway, and the area deprivation index (ADI) and family pediatrician in Italy. Unexposed children and adolescents could be selected as controls for matching more than once and might have contributed to both the unexposed and exposed cohorts if they were vaccinated later. This happened in 12% and 18% of cases, respectively in Italy and Norway. If an unvaccinated control subsequently received a vaccination, it was censored from the unexposed group along with its exposed match. In such cases, a new unvaccinated control was identified, allowing the newly vaccinated individual to be included in the exposed group (Appendix A—Panel A).

#### 2.3.2. Hybrid Immunity and Vaccine-Induced Immunity with Two Doses

Using the vaccination and SARS-CoV-2 infection registries, patients were classified as (i) having vaccine-induced immunity if they had recorded the receipt of the full two-dose COVID-19 vaccination schedule with a 21–50-day interval between doses without a record of previous infection and (ii) having hybrid immunity if they recorded a SARS-CoV-2 infection followed by a first dose of COVID-19 vaccine within 12 months [21].

We applied a lag time of up to 50 days between the two doses to align with established guidelines, which recommend delaying the second dose in children experiencing acute infection following their first vaccination [22]. Children with a documented SARS-CoV-2 infection between the two doses were excluded.

Children with vaccine-induced immunity were randomly selected and matched 1:1 based on matching variables described previously and the calendar date of achieved immunization (±15 days). The index date was defined as the date of the achievement of immunization (Appendix A—Panel B). This date corresponded to (i) the date of the second dose for individuals with vaccine-induced immunity, and (ii) the date of the first dose for individuals with hybrid immunity.

### 2.4. Follow-Up and Outcomes

Follow-up in each group began on day 7 for individuals who received two vaccine doses and on day 14 for those who received a single dose, starting from the index date. Follow-up continued until the earliest of the following events: a microbiologically confirmed SARS-CoV-2 infection (i.e., the outcome of interest), loss to follow-up (e.g., change of residence, change of family physician, or death), or the approval of the bivalent SARS-CoV-2 vaccines—on 14 September 2022, for Italy, and 31 December 2022, for Norway—whichever occurred first.

SARS-CoV-2 infection was defined as any record of a positive SARS-CoV-2 test as ascertained by the MSIS (PCR) supplemented with confirmed COVID-19 infections diagnosed in primary and secondary care (ICPC-2 code: R992 and ICD-10 code: U071) in Norway and the swab registry in Italy (rapid and PCR). Reinfection with SARS-CoV-2 was defined as a second positive test/confirmed diagnosis registered at least 90 days after the first, in accordance with Italian guidelines [23].

### 2.5. Covariates Definition

The sociodemographic and clinical characteristics of interest were sex; age class (children: 5–11 years; adolescents: 12–14 years); parental vaccination status against COVID-19 (for Norway) and ADI (for Italy); presence of baseline somatic comorbidities (Appendix A); the number of SARS-CoV-2 infections prior to the index date, and the time in months since the most recent SARS-CoV-2 infection recorded before the index date. Children were classified as (i) SARS-CoV-2-naive if they did not have any record of previous SARS-CoV-2 infection, (ii) recent SARS-CoV-2-recovered if an infection was recorded within the 12 months preceding the index date, and (iii) past SARS-CoV-2-recovered if an infection was recorded more than 12 months before the index date.

To account for baseline health and ongoing medical monitoring, we also considered healthcare utilization intensity as a proxy. This was measured by the number of outpatient visits (categorized as 0, 1–5, 6–10, ≥11) and number of hospitalizations (categorized as 0, 1, ≥2) in the year preceding the index date.

The ADI, computed at the census block level using data from the 2011 Italian Census, is a socioeconomic measure ranging from 1 (lowest deprivation) to 5 (highest deprivation). It was calculated based on five parameters: low education, unemployment, living in rented housing, overcrowded households, and single-parent families [24].

### 2.6. Statistical Analysis

Eligible and matched cohorts were described with frequency distributions and measures of central tendency. Standardized mean differences (SMDs) were calculated to assess balance between the groups, with an SMD of less than 0.1 considered indicative of adequate balance.

To investigate the effectiveness of (i) at least one dose of COVID-19 vaccines compared to unvaccinated individuals, and (ii) hybrid immunity compared to vaccine-induced immunity, we fit conditional Cox regression models to estimate hazard ratio (HR) and corresponding 95% confidence interval (95% CI), respectively, for each matched cohort. All models were adjusted for comorbidities and healthcare utilization intensity. The effectiveness of at least one dose of vaccination and hybrid immunity was calculated as [1 − (HR)] × 100. To account for the potential impact of previous infection status, analyses were stratified by SARS-CoV-2-naive, recent SARS-CoV-2-recovered, and past SARS-CoV-2-recovered individuals. In the latter two stratified analyses, we additionally adjusted for the time in days since the first infection. We did not perform multiple imputations as no data on exposure, outcomes, and confounders were missing.

SAS software, version 9.4 (SAS Institute, Cary, NC, USA), and STATA were used for the analyses, and two-tailed *p*-values of less than 0.05 were considered significant for all hypotheses.

### 2.7. Subgroup and Sensitivity Analysis

To assess the robustness of our results, we performed several stratified and sensitivity analyses considering factors that might influence vaccine efficacy. We (i) focused the analysis on individuals with previous infections during the Omicron wave, reflecting the predominant circulating variant in Norway and Italy in the immediate post-vaccination period [25]; (ii) stratified participants by age class at the time of vaccine regulatory approval in children and adolescents (i.e., January 2022 and November 2021 for children, September 2021 and May 2021 for adolescents, in Norway and Italy, respectively); (iii) censored follow-up data at the time of receipt of the second dose to account for potential confounding by the second dose; and (iv) considered infections recorded within the first 14 days after the index date as outcome. In Norway, vaccines were offered based on availability, resulting in most children/adolescents receiving the second dose several months after the first. To account for this, we considered a larger gap (up to 150 days) between the two doses in a sensitivity analysis. Lastly, in Norway, we employed a stricter definition of SARS-CoV-2 infection by considering a record of PCR positive test for SARS-CoV-2 as registered in MSIS only.

In Norway, the Regional Committee for Medical and Health Ethics of South/East Norway (no. 285687 on 15 September 2021) approved the study. The Norwegian Data Protection Services for research and the University of Oslo approved the Data Protection Impact Assessment—DPIA (no. 341884). In Italy, the access to the database was granted by the Internal Scientific Committee of Società Servizi Telematici Srl, the legal owner of the Pedianet network on 2 May 2023. We followed the Strengthening the Reporting of Observational Studies in Epidemiology (STROBE) guidelines.

## 3. Results

We included 626,537 and 38,938 children and adolescents from 5 to 14 years in Norway and Italy, respectively, who were eligible for COVID-19 vaccination upon regulatory approval dates (January 2022 and November 2021 for children, September 2021 and May 2021 for adolescents—Figure 1). Table 1 and Table 2 present the sociodemographic and clinical characteristics of the target populations in Norway and Italy. Both cohorts comprised approximately 48% males and 52% females. In both countries, the population included mainly children, with 70% and 82% of included individuals aged 5–11 years in Norway and Italy, respectively. Approximately 28% of individuals in both countries had pre-existing comorbidities (Appendix A). In Norway, 91.4% of parents were vaccinated against COVID-19. In Italy, the ADI distribution was relatively homogenous, with 18% of individuals residing in the lowest ADI category and 14% in the highest.

### 3.1. Vaccine-Induced Immunization with a Single Dose

The matched cohorts included 6933 children and 139,809 adolescent pairs in Norway, and 11,959 children and 3783 adolescent pairs in Italy. Compared to unvaccinated individuals, vaccinated individuals in both countries were more likely to be SARS-CoV-2-naive prior to the index date (98% vs. 89% in Norway, 90% vs. 82% in Italy, respectively, for vaccinated and unvaccinated individuals). Additionally, vaccinated individuals had a longer median time since the last infection (6.5 [IQR 4.7 to 10.3] in Norway and 9.6 [5.9 to 12.5] in Italy) than unvaccinated individuals (3.5 [0.9 to 7.1] in Norway and 2.7 [0.8 to 9.6] in Italy). In both countries, about 50% of individuals had 1–5 outpatient visits in the year prior to the index date.

The effectiveness of a single dose of the mRNA COVID-19 vaccine varied across countries and by prior infection status (Figure 2, Appendix A).

In Norway, vaccination did not significantly reduce the risk of SARS-CoV-2 infection among SARS-CoV-2-naïve individuals (HR: 1.05, 95% CI: 1.04–1.07), while a modest 8% reduction in risk was observed in Italy (HR: 0.92, 95% CI: 0.88–0.96). These results were consistent in children and adolescents. Among individuals with SARS-CoV-2 infection, one-dose vaccination significantly reduced the risk of reinfection in Norway (HR: 0.10, 95% CI: 0.08–0.13, and HR: 0.10, 95% CI: 0.05–0.20, for recent and past infected individuals respectively). However, no significant association was found in Italy (HR: 1.22, 95% CI: 0.83–1.80 and HR: 0.55, 95% CI: 0.27–1.11, for recent and past infected individuals, respectively). This pattern was consistent across both pediatric and adolescent cohorts. However, when infections within 14 days of the index date were included, statistical significance was reached for past infections also in Italy (HR: 0.45, 95% CI: 0.24–0.85). Finally, when censoring the follow-up period at the time of receipt of the second vaccine dose, results showed that individuals vaccinated with a single dose had a lower risk of having a SARS-CoV-2 infection than unvaccinated individuals in Italy (HR: 0.73, 95% CI: 0.66–0.82). In Norway, the findings were further confirmed when the outcome was based on positive tests from MSIS only (Appendix A). Given the small proportion of individuals with prior Omicron infections in both countries, sub-analyses specifically examining this group were not deemed feasible.

### 3.2. Hybrid Immunity Versus Vaccine-Induced Immunity with Two Doses

A total of 212 (gap 21–50 days) and 12,873 (gap 21–150 days) matched pairs were identified in Norway, and 498 matched pairs were identified in Italy. Our analysis revealed that hybrid immunity provided greater protection against SARS-CoV-2 infection compared to vaccine-induced immunity alone. In Norway, hybrid immunity was associated with a 26% risk reduction (HR: 0.74, 95% CI: 0.47–1.16) for shorter dosing intervals (21–50 days) and a 44% reduction (HR: 0.56, 95% CI: 0.52–0.60) for longer intervals (21–150 days). In Italy, hybrid immunity conferred an even greater risk reduction equal to 86% (HR: 0.14, 95% CI: 0.09–0.21) (Figure 3, Appendix A). These findings remained consistent when follow-up was censored at the second dose for individuals with hybrid immunity.

## 4. Discussion

This study contributes to the limited literature on the effectiveness of single-dose COVID-19 vaccination in children aged 5–11 years and adolescents aged 12–15 years, considering previous SARS-CoV-2 infection and its timing. The study also provides novel data on the effectiveness of hybrid immunization in children and adolescents in two distinct epidemiological settings: Norway, which experienced a lower infection burden, and Italy, where early pandemic waves led to widespread infection and severe disease [26]. A key finding is that in both Norway and Italy, children and adolescents with hybrid immunity exhibited stronger protection against reinfection compared to those with vaccine-induced immunity, with estimates ranging from 86% in Italy to 44% in Norway. Conversely, as expected, in SARS-CoV-2-naïve individuals, the protection against infection conferred by a single vaccine dose was limited, ranging from 0.92 in Italy to 1.05 in Norway. However, prior SARS-CoV-2 infection combined with at least one dose of vaccine significantly enhanced protection against reinfection compared to those who remained unvaccinated after infection in both countries.

Our results align with a large population-based case-control study conducted between December 2021 and March 2022, which analyzed data from approximately 700,000 individuals aged 12 years and older [27]. This study assessed the interaction between the timing of infection, circulating variants, and immunity, comparing protection against Omicron infection and severe COVID-19 conferred by previous infection (both Omicron and pre-Omicron), vaccination alone with Wuhan-like mRNA vaccines, and hybrid immunity. Among unvaccinated individuals, prior infection was associated with a 44% (95% CI: 38–48) reduction in the risk of reinfection with Omicron. However, this protection declined significantly over time, from 66% at three months to 30% at 11 months post-primary infection. These findings are consistent with previous research documenting a biphasic antibody decay, with a rapid decline within the first three months post-infection followed by a slower reduction from six months onward in pediatric populations [28]. Although immunological correlates of protection against SARS-CoV-2 reinfection remain undefined, evidence suggests that children with higher naturally acquired antibody titers have a lower risk of reinfection compared to older children and adolescents with lower titers [28,29].

Consistent with the literature, our study supports the greater durability and efficacy of hybrid immunity over vaccine-induced immunity alone. Similar findings have been reported in studies showing enhanced production of Omicron-specific neutralizing antibodies and stronger cellular responses in SARS-CoV-2-recovered children who had been infected with pre-Omicron variants, compared to naïve children. Rothoeft et al. [30] investigated the humoral and cellular immune responses in 72 children and adolescents following SARS-CoV-2 infection and vaccination with tozinameran. Their study included individuals with natural infection and those with hybrid immunity assessed 3–26 months post-infection. The findings demonstrated that hybrid immunity was associated with significantly higher antibody titers and a greater likelihood of a cellular immune response compared to natural immunity alone. These immunological findings are in line with ours, showing that hybrid immunity, even if differently defined, provides higher protection from reinfection. Similarly, other studies have reported higher production of Omicron-specific neutralizing antibodies and a stronger cellular response following one or two doses of COVID-19 vaccine in SARS-CoV-2-recovered children who had been infected with pre-Omicron variants, compared to SARS-CoV-2-naïve children [31,32,33]. These findings support the immunological rationale for the greater effectiveness of hybrid immunity against Omicron variants. Continued surveillance is crucial to monitor the evolving landscape of COVID-19, including the impact of emerging variants on vaccine effectiveness as well as the long-term durability of both vaccine-induced and hybrid immunity in this population. Ongoing research on vaccine effectiveness is therefore critical to inform immunization strategies and to protect the health of children and adolescents—both by preventing acute morbidity and by reducing the risk of long-term sequelae associated with SARS-CoV-2 infection [34,35,36].

### 4.1. Strengths and Limitations for the Norwegian Database

The study is based on multiple, nation-wide health registry data in Norway. The mandatory registration of COVID-19 vaccination in SYSVAK minimizes the risk of misclassification of vaccine uptake. The comprehensive availability of clinical diagnoses from primary and secondary healthcare enabled us to adequately identify comorbidities and outcomes of interest.

This study also has some limitations that need consideration when interpreting the results. Vaccine doses given abroad need manual retrospective registration in SYSVAK, leading to uncertainty about the number of individuals who received vaccines overseas without official records in Norway. However, this risk is considered minimal, given the necessity of vaccine registration during the study period.

### 4.2. Strengths and Limitations for the Pedianet Database in the Veneto Region, Italy

For this analysis, we used the Pedianet database, which had previously been confirmed representative of Italy, in particular the Veneto Region, covering 10% of the regional paediatric population [37]. Moreover, given the longevity and the administrative and surveillance use of the immunization registry in the Veneto region, vaccination data are of high standard.

Our study also had several limitations. The study is of a retrospective nature, which, as with any observational study, does not allow for the elimination of the possibility that patients receiving vaccination differ from those who did not receive it for some unmeasured features that the pediatrician did not report in the medical records. The limited sample size, mostly within the recent and past SARS-CoV-2-recovered subject groups, may have resulted in low statistical power. However, the median time since recent infection prior to the index date was 4.3 months for recently recovered subjects, whereas it was 13.5 months for past-recovered subjects. This temporal disparity could potentially explain the observed null effectiveness among recently recovered children and the positive effectiveness among those with past recovery, aligning with current booster guidelines in Italy. In addition, misclassification might have affected the study. Some children who had SARS-CoV-2 infection might not have been traced, and some children with asymptomatic infections might not have undergone testing. Furthermore, we did not analyze the VE by vaccine type because very few children and adolescents were vaccinated with elasomeran [38]. However, a meta-analysis on adults found that both mRNA-based vaccines hold a similar VE [39].

## 5. Conclusions

Hybrid immunization is shown to provide enhanced protection against SARS-CoV-2 reinfection in both children and adolescents, offering greater defense against various VOCs. These findings emphasize the importance of staying up-to-date with seasonal COVID-19 vaccinations, as this strategy plays a crucial role in reducing the risk of reinfection in children. Moreover, children play a significant role in the transmission and shedding of respiratory viruses, including SARS-CoV-2 [40].

Finally, the research highlights variations in the magnitude of vaccine effectiveness across different countries, underscoring the need for context-specific studies on vaccine efficacy, as the local research environment can influence the generalizability of results.

## Figures and Tables

**Figure 1 vaccines-13-00698-f001:**
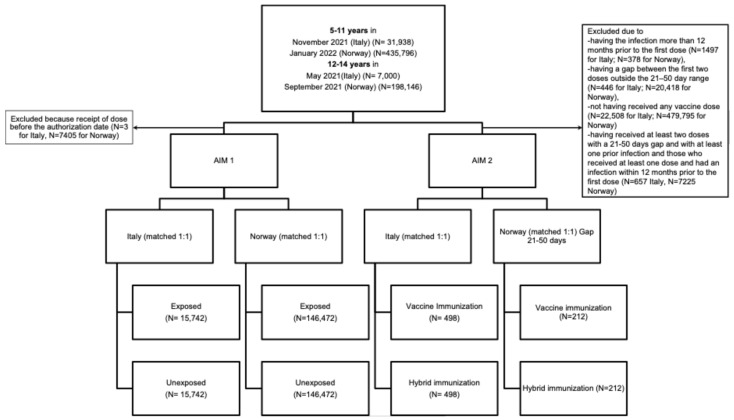
Flow chart of included individuals by country and by study aim.

**Figure 2 vaccines-13-00698-f002:**
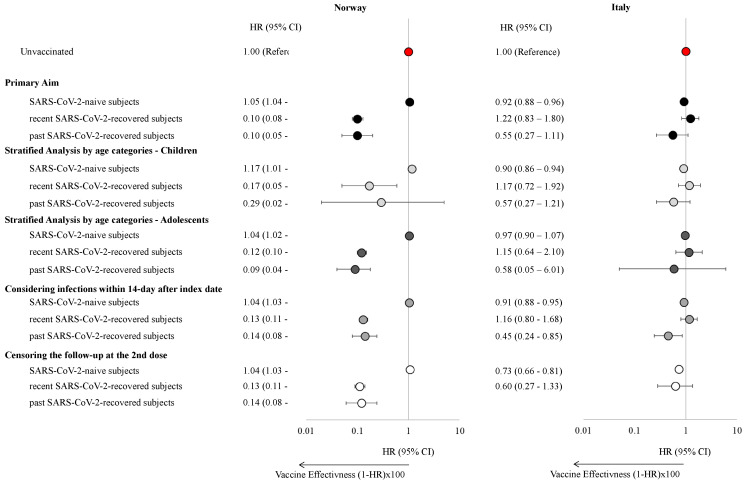
Hazard ratios with 95% confidence intervals of experiencing a SARS-CoV-2 infection stratified by the presence of a previous SARS-CoV-2 infection, age class, and by varying the censoring and the outcome eligible period in children and adolescents in Norway and the Veneto Region, Italy.

**Figure 3 vaccines-13-00698-f003:**
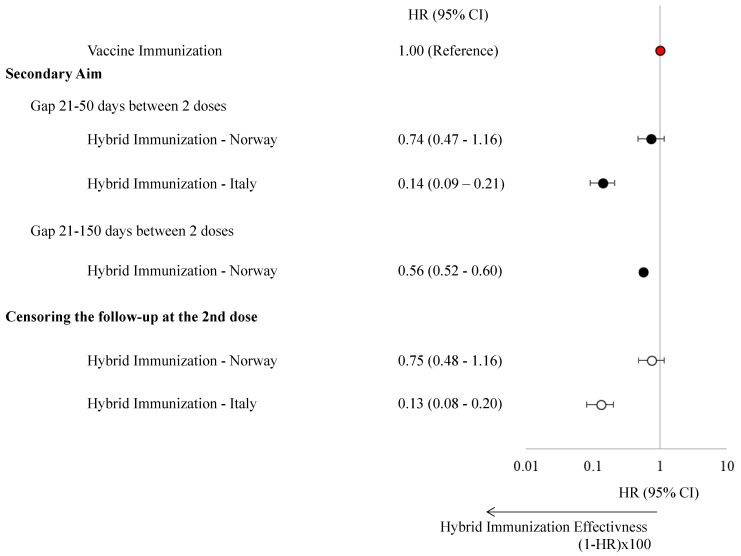
Hazard ratios with 95% confidence intervals of experiencing a SARS-CoV-2 infection stratified by vaccine-induced immunization and hybrid immunization in children and adolescents in Norway and the Veneto Region, Italy.

**Table 1 vaccines-13-00698-t001:** (**A**) Target population and matched cohorts for the primary aim. Norway, N = 626,537 children and adolescents. (**B**) Matched cohorts for the second aim. Norway.

A						
			Primary Aim			
	Overall		Exposed to the 1st Dose of Vax	Unexposed		SMDs
	(N = 626,537)		(N = 146,472)	(N = 146,472)		
Sex—N (%)						M.V.
Female	304,560 (48.6)		71,547 (48.8)	71,547 (48.8)		
Male	321,977 (51.4)		75,195 (51.2)	75,195 (51.2)		
Age class—N (%)						M.V.
Children (5–11 yr.)	435,546 (69.5)		6933 (4.7)	6933 (4.7)		
Adolescents (12–15 yr.)	190,991 (30.5)		139,809 (95.3)	139,809 (95.3)		
Parental vaccination						
Yes	582,943 (93.0)		146,007 (99.5)	146,007 (99.5)		M.V.
No	43,594 (7.0)		735 (0.5)	735 (0.5)		
Number of outpatient visits						
0			18,258 (12.4)	19,987 (13.6)		−0.036
1–5			82,545 (56.3)	80,892 (55.1)		0.024
6–10			31,167 (21.2)	31,361 (21.3)		−0.002
≥11			14,772 (10.7)	14,502 (9.9)		0.026
Number of hospitalizations						
0			142,004 (96.8)	142,324 (97.0)		−0.011
1			3833 (2.6)	3713 (2.5)		0.006
≥2			905 (0.6)	705 (0.5)		0.014
Previous comorbidities in the previous 10 years—Yes (%) ∫	182,481 (29.1)		46,272 (31.5)	45,237 (30.8)		-
Previous comorbidities in the previous 5 year—Yes (%) ∫	134,639 (21.5)		32,617 (22.2)	31,995 (21.8)		-
Time Since Last Infection in mo.—Median [IQR] ¥			6.5 (4.7–10.3)	3.4 (0.9–7.0)		
Children with Previous Infection—N (%)						
No—SARS-CoV-2-naïve			143,822 (98.0)	133,921 (91.3)		0.301
Yes—Recent SARS-CoV-2 Infection			2542 (1.7)	12,424 (8.5)		−0.313
Yes—Past SARS-CoV-2 Infection			378 (0.3)	397 (0.3)		−0.003
**B**						
	**Gap 21–50**			**Gap 21–150**		
	**Vaccine Immunization**	**Hybrid Immunization**	**SMDs**	**Vaccine Immunization**	**Hybrid Immunization**	**SMDs**
	**(N = 212)**	**(N = 212)**		**(N = 12,873)**	**(N = 12,873)**	
Sex—N (%)			M.V.			M.V.
Female	97 (45.8)	97 (45.8)		6588 (51.2)	6588 (51.2)	
Male	115 (54.2)	115 (54.2)		6285 (48.8)	6285 (48.8)	
Age class—N (%)			M.V.			M.V.
Children (5–11 yr.)	47 (22.2)	47 (22.2)		710 (5.5)	710 (5.5)	
Adolescents (12–15 yr.)	165 (77.8)	165 (77.8)		12,163 (94.5)	12,163 (94.5)	
Parental Vaccination						
Yes	211 (99.5)	212 (99.5)	M.V.	12,851 (99.8)	12,851 (99.8)	M.V.
No	<5 (0.5)	<5 (0.5)		22 (0.2)	22 (0.2)	
Number of Hospitalizations						
0	192 (90.6)	208 (98.1)	−0.329	12,288 (95.5)	12,465 (96.8)	−0.068
1	13 (6.1)	4 (1.9)	0.216	431 (3.4)	312 (2.4)	0.060
≥2	7 (3.3)	0	0.261	154 (1.2)	96 (0.8)	0.040
Number of Outpatient Visits						
0	36 (17.0)	7 (3.3)	0.466	1517 (11.8)	633 (4.9)	0.251
1–5	96 (45.3)	92 (43.4)	0.038	6775 (52.6)	6017 (46.7)	0.118
6–10	32 (15.1)	76 (35.9)	−0.491	2841 (22.1)	4011 (31.2)	−0.207
≥11	48 (22.6)	37 (17.5)	0.128	1740 (13.5)	2212 (17.2)	−0.103
Previous Comorbidities in the Previous 10 years—Yes (%) ∫	82 (38.7)	77 (36.3)	-	4522 (35.1)	4070 (31.6)	-
Previous Comorbidities in the Previous 5 year—Yes (%) ∫	72 (34.0)	63 (29.7)	-	3370 (26.2)	3063 (23.8)	-

SMD: standardized mean difference; M.V.: matching variable; ∫ Previous comorbidities reported in Appendix A; ¥ Evaluated only among children with a previous infection over the index date.

**Table 2 vaccines-13-00698-t002:** Target population and matched cohorts for the primary aim. Italy, N = 38,938 children and adolescents.

		Primary Aim			Secondary Aim		
	Overall	1-Dose Vaccination	Unvaccinated	SMDs	Vaccine Immunization	Hybrid Immunization	SMDs
	(N = 38,938)	(N = 15,742)	(N = 15,742)		(N = 498)	(N = 498)	
Sex—N (%)				M.V.			M.V.
Female	18,774 (48.2)	7606 (48.3)	7606 (48.3)		232 (46.2)	232 (46.2)	
Male	20,164 (51.8)	8136 (51.7)	8136 (51.7)		270 (53.8)	270 (53.8)	
Age Class—N (%)				M.V.			M.V.
Children (5–11 years)	31,938 (82)	11,959 (76)	11,959 (76)		317 (63.1)	317 (63.1)	
Adolescents (12–15 years)	7000 (18)	3783 (24)	3783 (24)		185 (36.9)	185 (36.9)	
Area Deprivation Index—N (%)				M.V.			M.V.
1-Lowest	7183 (18.5)	3127 (19.9)	3127 (19.9)		92 (18.3)	92 (18.3)	
2	6998 (18)	2983 (19)	2983 (19)		116 (23.1)	116 (23.1)	
3	6447 (16.6)	2561 (16.3)	2561 (16.3)		83 (16.5)	83 (16.5)	
4	5868 (15.1)	2207 (14)	2207 (14)		69 (13.8)	69 (13.8)	
5-Highest	5554 (14.3)	2077 (13.2)	2077 (13.2)		73 (14.5)	73 (14.5)	
Missing	6888 (17.7)	2787 (17.7)	2787 (17.7)		69 (13.7)	69 (13.7)	
Number of Outpatient Visits							
0	-	4851 (30.8)	4667 (29.7)	0.023	431 (86.6)	410 (82.3)	0.119
1–5	-	7711 (48.5)	7883 (50.1)	−0.032	44 (8.8)	48 (9.6)	−0.027
6–10	-	2396 (15.2)	2363 (15)	0.005	23 (4.6)	40 (8.1)	−0.144
≥11	-	784 (5)	829 (5.3)	−0.013	-	-	
Previous Comorbidities—Yes (%) ∫	10,641(27.3)	4265 (27)	4289 (27.3)	−0.007	168 (33.5)	163 (32.5)	0.021
Time Since Last Infection in months—Median [IQR] ¥	-	9.6 (5.9–12.5)	2.7 (0.8–9.6)	-		8.2 (5.4–9.8)	-
Children with Previous Infection—N (%)	-						
No—SARS-CoV-2-naïve	-	14,160 (89.9)	12,878 (81.8)	0.23	498 (100)	-	-
Yes—Recent SARS-CoV-2 Infection	-	1113 (7.1)	2413 (15.3)	−0.26	-	502 (100)	-
Yes—Past SARS-CoV-2 Infection	-	469 (3)	451 (2.9)	0.006	-	-	-

SMD: standardized mean difference; M.V.: matching variable; ∫ Previous comorbidities reported in Appendix A; ¥ Evaluated only among children with a previous infection over the index date.

## Data Availability

The data underlying this article are not publicly available. For Italian data, deidentified data could be shared upon reasonable request to the corresponding author and approval of the Internal Scientific Committee of Società Servizi Telematici Srl, the legal owner of Pedianet.

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
