# Peer review of "Impact of Prior SARS-CoV-2 Infection on COVID-19 Vaccine Effectiveness in Children and Adolescents in Norway and Italy"

_vaccines, 2025, doi:10.3390/vaccines13070698_

Round 1

Reviewer 1 Report

Comments and Suggestions for Authors

Thank you for the opportunity to review the manuscript ID: vaccines-3637933. This manuscript aimed to assess the effectiveness of a single dose of COVID-19 vaccine, and compare the effectiveness of hybrid immunity versus complete vaccination in the prevention of SARS-CoV-2 infection in a cohort of children and adolescents aged 5-14 years in Norway and the Veneto Region, Italy.     

Overall, this manuscript follows the appropriate structure, but some issues are evident that need to be addressed.        

Comments: 

Abstract:
- Lines 34-36: Check where the results mentioned in this sentence are presented in this manuscript (I quote `A single dose of the vaccine did not reduce the risk of infection in Norway for SARS-CoV-2-naive individuals (HR=1.05; 95%CI=0.96–1.14), whereas it reduced risk by 8% in Italy (HR=0.92; 95%CI=0.85–0.99).`). Correct this. 
- Lines 36-38: Check where the results mentioned in this sentence are presented in this manuscript (I quote `In individuals with prior-recent infection, vaccination reduced reinfection risk in Norway (HR=0.10; 95%CI=0.08–0.12), but not in Italy (HR=1.22; 95%CI=0.99–1.51) vs no vaccination.`). Correct this.        
- Lines 38-40: In this sentence, specify whether the reduction in reinfection was significant (I quote `Among those with prior infection, vaccination reduced reinfection risk in both Norway (HR=0.10; 95%CI=0.05–0.20), and Italy (HR=0.55; 95%CI=0.27–1.11).`). Correct this.       
- Lines 40-43: Check where the results mentioned in this sentence are presented in this manuscript (I quote `Hybrid immunity provided greater protection than vaccine-induced immunity, with a 26% risk reduction in Norway (HR=0.74; 95%CI=0.71–0.77) and 86% in Italy (HR=0.14; 95%CI=0.11–0.17) against (re-)infection.`). Correct this.  

Methods section:   
- Instead of Figure S1, which is unsatisfactorily clear and does not contain a precise Legend, provide a Flow chart for this study.           
- Lines 148-150: Explain whether the procedure presented in this sentence (I quote `Unexposed children and adolescents could be selected as controls for matching more than once and might have contributed to both the unexposed and exposed cohorts if they were vaccinated at a later date.`), that is, the data obtained according to this criterion, might have influenced the results of this study, either in terms of underestimating or overestimating of the HR. 
If they have an effect, please state so in the subsection `Limitations of this study`.         
- Line 217: Define the level at which statistical significance is considered.                  

Results section:
- Table 1A: Values ​​for the categories `Yes - Recent SARS-CoV-2-infection' and `Yes - Past SARS-CoV-2-infection' in the variables `Children with previous Infection - N (%)' and `Children with previous Infection and confirmed COVID-19 diagnoses - N (%)' should be checked, explained and corrected.         

Conclusions section:   
- Lines 415-418: Move this sentence to the appropriate place in the Discussion section. Rationale: In the Conclusions section, one should emphasize one's own results, without citing other authors' studies.      

Author Response

We thank the Reviewer for their valuable comments on our manuscript. Please find our detailed responses attached.

Reviewer 2 Report

Comments and Suggestions for Authors

The manuscript "Impact of Prior SARS-CoV-2 Infection on COVID-19 Vaccine Effectiveness in Children and Adolescents in Norway and Italy" by Elisa Barbieri et al., uses a retrospective cohort to study the role of prior SARS-CoV-2 infection in vaccine-induced immunity and protection against reinfection.
I agree with the authors that this manuscript provides valuable insights about the role of hybrid immunization in enhancing protection against SARS-CoV-2 reinfection in both children and adolescents.

Here are my minor suggestions:

1. I am concerned about the relevance of the dataset. Could you clarify the reasoning behind comparing data from the Norwegian database with the Pedianet database? 

2. What would the results look like if the data were stratified by age group (children versus adolescents)?

Author Response

(The authors gave the same response as above.)

Reviewer 3 Report

Comments and Suggestions for Authors

Thank you for submitting an interesting and useful paper.

This study clearly concludes the significance of the vaccinating children and adolescents who have had COVID-19 in the past. The large number of study participants is also a strength of this study. The paper is also well written.

I would like to ask a question about the methodology. Could you mention to what extent the participants in this study had underlying diseases? In particular, given the exclusion criteria, were there any cases of immunocompromised patients?

Author Response

(The authors gave the same response as above.)

Round 2

Reviewer 1 Report

Comments and Suggestions for Authors

Thank you for the opportunity to re-review the manuscript ID: vaccines-3637933.    

The authors made some corrections in this manuscript. Thanks to the authors.
However, some issues were not addressed, including:     

Abstract:
- Lines 34-37: Check where the results mentioned in this sentence are presented in this manuscript (I quote `A single dose of the vaccine did not reduce the risk of infection among SARS-CoV-2–naive individuals in Norway (HR: 1.05; 95% CI: 0.96–1.14), whereas it was associated with an 8% risk reduction in Italy (HR: 0.92; 95% CI: 0.85–0.99).`). Check and mark (in which Table or Figure of this manuscript?). Correct this.           
- Lines 39-42: Check where the results mentioned in this sentence are presented in this manuscript (I quote `Among individuals with a recent prior infection (within 12 months), vaccination was associated with a reduced risk of reinfection in Norway (HR: 0.10; 95% CI: 0.08–0.12), but not in Italy (HR: 1.22; 95% CI: 0.99–1.51), compared to no vaccination.`). Check and mark (in which Table or Figure of this manuscript?). Correct this.                      
- Lines 48-51: Check where the results mentioned in this sentence are presented in this manuscript (I quote `Hybrid immunity provided greater protection against (re-)infection compared to vaccine-induced immunity alone, with a 26% risk reduction observed in Norway (HR=0.74; 95%CI=0.71–0.77) and an 86% reduction in Italy (HR=0.14; 95%CI=0.11–0.17).`). Check and mark (in which Table or Figure of this manuscript?). Correct this.            

Supplementary Figure S1 in the revised version of this manuscript does not represent `Flow chart of this study`, but only `Definition of the study cohorts` (as that Figure is titled). Correct this, in such a way that the flow of these 2 studies is presented in the Figure S1.     

Author Response

Responses to R1 are attached

Round 3

Reviewer 1 Report

Comments and Suggestions for Authors

Thanks for the opportunity to re-review this manuscript. However, none of my comments were addressed. 

  The Authors did not provide answers to my first three comments. Namely, I did not ask for the illustrations to be cited in the Abstract as the Authors wrongly claimed in their response. In fact I pointed out that these results that are listed in the Abstract (nota bene: first three comments I made from the previous revision round) are not mentioned anywhere in the manuscript, and asked that this is corrected with appropriate linking to the corresponding illustration that shows these results - because if you actually check you will see that the results mentioned in the abstract for all three comments that I stated do not match those in the text of the manuscript nor in the Figures 1 and 2. Some even completely wrongly present results as significant - opposed to that on the Figures or in the text.    In the Supplement material, Figure 2 on sheet 2 is not named as stated in the Response letter. But more importantly, none of these two Figures actually show the flow chart of the study. Specifically, a flow chart depicts specifics on everything that was done shown in a timeline manner with details on e.g. inclusion/exclusion criteria, matching and randomization details (with numbers of considered population, included, excluded, ...), then steps when everything relevant to the study was done (data collection, follow-up, numbers, etc) - as appropriate for the conducted study.         

Author Response

Padova, June 12, 2024

Dear Reviewer,

Apologies for the misunderstanding. We have now amended the typos in the abstract which caused inconsistency with the confidence intervals reported in Figure 1 and 2 (now Figure 2 and 3) in the text. Also, we have now implemented Figure 1 depicting a flow chart of individuals included by study aim and by country as suggested.

Best regards,

Costanza Di Chiara, on behalf of all the authors